# Use of DNA Barcoding Combined with PCR-SFLP to Authenticate Species in Bison Meat Products

**DOI:** 10.3390/foods10020347

**Published:** 2021-02-06

**Authors:** Zerika M. Scales, Elif Narbay, Rosalee S. Hellberg

**Affiliations:** Food Science Program, Schmid College of Science and Technology, Chapman University, 1 University Drive, Orange, CA 92866, USA; scales@chapman.edu (Z.M.S.); narbay@chapman.edu (E.N.)

**Keywords:** bison, cattle, DNA barcoding, hybrids, PCR-RFLP, PCR-SFLP, species identification

## Abstract

American bison (*Bison bison*) meat is susceptible to species mislabeling due to its high value and similar appearance to meat from domestic cattle (*Bos taurus*). DNA barcoding is commonly used to identify animal species. However, as a result of the historical hybridization of American bison and domestic cattle, additional genetic testing is required for species confirmation. The objective of this study was to perform a market survey of bison meat products and verify the species using DNA barcoding combined with polymerase chain reaction-satellite fragment length polymorphism (PCR-SFLP). Bison products (*n* = 45) were purchased from a variety of retailers. Samples that were positive for domestic cattle with DNA barcoding were further analyzed with PCR-SFLP. DNA barcoding identified bison in 41 products, red deer (*Cervus elaphus*) in one product, and domestic cattle in three products. PCR-SFLP confirmed the identification of domestic cattle in two samples, while the third sample was identified as bison with ancestral cattle DNA. Overall, mislabeling was detected in 3 of the 45 samples (6.7%). This study revealed that additional DNA testing of species that have undergone historical hybridization provides improved identification results compared to DNA barcoding alone.

## 1. Introduction

American bison (*Bison bison*) once flourished in North America, numbering in the tens of millions [1,2]. However, great numbers of bison were slaughtered during the peak of the hide trade in the late 1800s, and the species was driven to near extinction. By the early 1900s, the remaining bison survived as small herds on five private ranches and within a small wild herd in Yellowstone National Park, which had less than 25 animals in 1902 [1,3]. Bison on the remaining ranches were crossbred with domestic cattle (*Bos taurus*) in an attempt to improve the traits of cattle, including meat quality, quantity, hardiness, feed efficiency, and disease resistance [4,5,6]. Although the crossbreeding events were discontinued, they resulted in the incorporation of cattle DNA into American bison populations. In 1905, the American Bison Society was formed, and its lobbying efforts led to the creation of several public conservation herds within the United States [7,8]. Currently, there are approximately 400,000 bison in commercial herds in North America and some 31,000 bison managed within conservation herds [9,10]. The International Union for Conservation of Nature (IUCN) Red List considers American bison to be “Near Threatened”; however, the population is considered stable and there is a harvest management plan in place [9].

The North American bison industry experienced steady growth from 2010 to 2020, with sales of bison meat in restaurants and retail stores reaching USD 350 million per year [11]. The majority of bison producers (80%) sell directly to consumers and restaurants rather than grocery stores [12]. Bison sales fell in 2020 due to restaurant closures associated with the coronavirus pandemic; however, the industry is receiving support from the US Department of Agriculture (USDA), which agreed to purchase USD 17 million (~1.1 million kg) of bison meat from processors. Bison, which is sometimes mistakenly referred to as “buffalo,” is low in fat, calories, and cholesterol and contains high amounts of protein, iron, and vitamin B-12 [11]. Bison spend the majority of their lives on grasslands, with little or no time in the feedlot, and they are generally not given antibiotics or growth hormones. Bison meat is highly valued compared to beef from domestic cattle. For example, the average direct-to-consumer price for ground lean bison in March 2020 was USD 30.14/kg [13] compared to an average retail price of USD 12.40/kg for lean and extra lean ground beef [14]. During the same time period, the average direct-to-consumer price for bison ribeye steaks was USD 80.90/kg [13], which is almost five times that of the average retail price of beef steaks at USD 16.89/kg [14]. These price differences combined with the similar appearance of meat from bison and domestic cattle introduce the potential for intentional mislabeling for the purpose of economic gain.

Mislabeling species of meat products is commonly detected using DNA-based techniques, such as DNA barcoding and real-time polymerase chain reaction (real-time PCR) [15,16]. DNA barcoding is a widely used sequencing method in which universal primers target a short, standardized genetic region for the identification of species [17]. In animals, the most commonly used region is a ~650 base-pair (bp) fragment of the mitochondrial gene coding for cytochrome *c* oxidase subunit I (COI). Previous US market surveys using DNA-based techniques such as DNA barcoding or real-time polymerase chain reaction (PCR) have reported the identification of cattle in commercial bison or buffalo products [18,19,20]. For example, a market survey conducted on game meats sold in the US reported that two products labeled as “stewed bison meat” and “rib-eye bison steak” were identified as domestic cattle with DNA barcoding [19]. Another US market survey detected a mixture of beef, pork, and horse in a product labeled as “ground bison” using real-time PCR [18].

A shortcoming of previous market surveys involving bison products was that the analytical methods used for the detection of bison were based on mitochondrial DNA (mtDNA), which is inherited through the maternal line [19]. This is problematic when testing for the presence of bison because historical crossbreeding of the two species was reliant on breeding male bison with female cattle, and backcrosses of male bison with female offspring [6]. Although the cross-breeding programs were halted over a century ago, some American bison populations still carry ancestral cattle DNA, with an average of 13.9% mtDNA cattle ancestry and 0.6% autosomal cattle ancestry across 22 herds studied [6]. Additionally, according to US regulations, the term bison may refer to American bison or cattalo, which is a result of direct crossbreeding between American bison and domestic cattle (Exotic Animals and Horses, 9 Code of Federal Regulations (CFR) § 352). As a result, DNA-based testing of bison products has the potential to give a false-positive result for cattle due to the presence of ancestral cattle DNA in bison or the use of cattalo in a bison product. However, previous studies reporting the detection of cattle DNA in bison products did not perform additional testing to verify the identity of the product. A method that could be used to confirm the species in these situations is polymerase chain reaction (PCR)-satellite fragment length polymorphism (SFLP), which is a variation of PCR-restriction fragment length polymorphism (RFLP) that targets centromeric satellite DNA. A PCR-SFLP method was previously developed for the differentiation of bovine species, including animals of hybrid origin such as bison and cattle [21]; however, it has not been used to verify species labeling of bison products.

The objective of this study was to perform a market survey of bison meat products and verify the species using a combination of molecular methods. DNA barcoding was used as an initial test for species and any bison products that were identified as domestic cattle were then analyzed using PCR-SFLP to verify the species.

## 2. Materials and Methods

### 2.1. Sample Collection

A total of 45 unique products advertised as containing bison or buffalo were purchased for use in this study (Table 1). The products included uncooked burger patties/ground meat (*n* = 13), cooked burgers (*n* = 21), uncooked whole cuts/steaks (*n* = 10), and hot dogs (*n* = 1). The products were purchased from 35 different retail outlets online and in Orange and Los Angeles Counties, CA, USA. The average price among the products collected was USD 14.99/product, with price ranges of USD 9.58 to USD 23.00 for cooked burgers, USD 5.89 to USD 16.99 for uncooked burger patties/ground meat, USD 8.99 to USD 30.00 for uncooked whole cuts/steaks, and USD 20.00 for the hot dog package. Following collection, all products were stored at −20 °C.

### 2.2. DNA Extraction

DNA extraction was carried out using the DNeasy Blood and Tissue Kit (Qiagen, Valencia, CA, USA), spin-column protocol, according to the manufacturer’s instructions. Samples were thawed at 4 °C overnight prior to testing. Approximately ~25 mg of tissue from each sample was removed from the interior portion of each meat product using a sterile scalpel and forceps. The tissue samples were placed directly into a 1.5 mL microcentrifuge tube for use in DNA extraction. Ground samples were placed in a 7-oz Whirl-Pak bag (Nasco, Salida, CA, USA) and the bag was massaged by hand for 60 s to obtain a homogenous mixture prior to tissue sampling [22]. Tissue samples were lysed for 3 h at 56 °C at 300 rpm in a thermomixer C (Eppendorf, Hauppauge, NY, USA) and DNA was eluted in 100 µL of preheated (37 °C) AE buffer (Qiagen, Valencia, CA, USA). Each set of DNA extractions included a reagent blank negative control with no tissue added.

### 2.3. DNA Barcoding 

A 658-bp region of the gene coding for COI was amplified using the mammalian primer cocktails (Table 2) described in Ivanova et al. [23]. Each PCR tube contains the following: 12.5 µL HotStar Taq Master Mix (Qiagen, Valencia, CA, USA), 10 µL molecular grade water, 0.25 µL forward primer cocktail (10 µM; Table 2), 0.25 µL reverse primer cocktail (10 µM; Table 2), and 2 µL template DNA. A non-template control (NTC) with sterile water in place of DNA was included in each PCR run. Thermal cycling was carried out using a Mastercycler nexus gradient thermal cycler (Eppendorf, Hauppauge, NY, USA) with the following cycling conditions: 95 °C for 15 min; 5 cycles of 94 °C for 30 s, 50 °C for 40 s, and 72 °C for 1 min; 35 cycles of 94 °C for 30 s, 55 °C for 40 s, and 72 °C for 1 min; and a final extension step at 72 °C for 10 min [23].

PCR products were confirmed with gel electrophoresis by loading 4 µl of PCR product and 10 µl E-Gel^TM^ 1 Kb Plus DNA ladder (Invitrogen, Carlsbad, CA, USA) on 2.0% E-gels (Invitrogen, Carlsbad, CA, USA). The gels were run for 15 min on an E-Gel iBase Power System (Invitrogen, Carlsbad, CA, USA) [24]. The results were visualized with Foto/Analyst Express (Fotodyne, Hartland, WI, USA) in combination with Transilluminator FBDLR-88 (Fisher Scientific, Waltham, MA, USA) and PCIMAGE (version 5.0.0.0 Fotodyne, Hartland, WI, USA). The PCR products underwent purification with ExoSAP-IT (Affymetrix, Santa Clara, CA, USA) following the manufacturer’s instructions. DNA sequencing was carried out at the GenScript facility (Piscataway, NJ, USA). Samples were sequenced bidirectionally with M13 tails (Table 2) using a BigDye Terminator v3.1 Cycle Sequencing Kit (Life Technologies, Carlsbad, CA, USA) and a 3730xl Genetic Analyzer (Life Technologies, Carlsbad, CA, USA).

### 2.4. Sequence Analysis 

Raw sequencing data were assembled and edited with Geneious R7 (Biomatters Ltd., Auckland, New Zealand). The resulting consensus sequences underwent alignment with multiple sequence comparison by Log-Expectation (MUSCLE) using the default settings in Geneious R7. All sequences were trimmed to the standard 658-bp COI DNA barcode region. The length, % high-quality bases (HQ%), and ambiguities for each consensus sequence were recorded. The consensus sequences were searched against the public barcode records in the Barcode of Life Data Systems (BOLD) Identification System for COI. The species matches with the greatest genetic similarity to the query sequence were recorded. 

### 2.5. PCR-SFLP 

Samples that were identified as domestic cattle after DNA barcoding underwent further testing with PCR-SFLP targeting the satellite 1.711b amplicon, followed by digestion with TaqI [21]. Each PCR tube contained 8.5 µL molecular grade water, 12.5 µL HotStarTaq Master Mix (2×), 1 µL of each satellite 1.711b primer (50 ng; Table 2), and 2 µL template DNA, for a total volume of 25 µL. Thermal cycling was carried out using a Mastercycler nexus gradient thermal cycler (Eppendorf, Hauppauge, NY, USA) with the following cycling conditions: HotStarTaq activation for 15 min at 95 °C; 25 cycles of 15 s at 92 °C, 30 s at 38 °C, and 45 s at 72 °C; followed by a final extension for 5 min at 72 °C. PCR products next underwent a restriction digest with TaqI (Fisher Scientific, Hanover Park, IL, USA). Each restriction digest contained 10 µL PCR product, 2 µL 10X buffer R, 18 µL nuclease-free water, and 1 µL (10 U) of restriction endonuclease (TaqI). Restriction digests were carried out for 3 h at 65 °C using a Mastercycler nexus gradient thermal cycler. The PCR-SFLP products were separated with gel electrophoresis using the settings described above for DNA barcoding, with the exception that loading volumes of 20 µL were used and the gels were run for 30 min. Each PCR-SFLP assay included a non-template negative control, a positive bison DNA control verified through DNA barcoding (as described above) and a positive domestic cattle DNA control verified with the Tetraplex Real-Time PCR AllHorseTM Assay (Microsynth AG, Balgach, Switzerland), according to the manufacturer’s instructions.

## 3. Results

### 3.1. DNA Barcoding 

All 45 samples were successfully amplified and sequenced with the COI DNA barcode. The consensus sequences are available in Appendix A. High-quality sequences were obtained, with an average length of 655 ± 6 bp, an average % high quality (HQ%) bases of 96.2 ± 7.5% and average percent ambiguities of 0.05 ± 0.16%. All sequences were >500 bp and had less than 2% ambiguities. The samples were all identified at the species level using BOLD with ≥99.7% genetic similarity to the top species match. The majority of samples (*n* = 41) showed a top species match to both American bison (*B. bison*) and steppe bison (*Bison priscus*). However, steppe bison became extinct about 10,000 years ago, at the end of the last Ice Age [25]. Therefore, this species identification was ruled out and the samples were determined to be American bison. The remaining four samples were identified as a species other than bison—one sample (Z021) was identified as red deer (*Cervus elaphus*) with 100% genetic similarity and three samples (Z003, Z011, Z014) were identified as domestic cattle (*B. taurus*) with 99.7–100% genetic similarity.

### 3.2. PCR-SFLP

The three samples that tested positive for domestic cattle with DNA barcoding were subjected to confirmatory testing with PCR-SFLP using the satellite 1.711b genetic marker combined with *Taq*I restriction enzyme (Figure 1). The expected result for bison was a single band at 822 bp, while the expected result for domestic cattle was three bands at 250 bp, 552 bp, and 809 bp [21]. As shown in Table 3, the results for sample Z011 and the positive bison DNA control were consistent with the expected result for bison. On the other hand, the results for samples Z003, Z014, and the positive cattle DNA control were all consistent with the expected result for domestic cattle. Overall, the combined results of DNA barcoding and PCR-SFLP indicated that among the three samples that tested positive for domestic cattle with DNA barcoding, two were domestic cattle (Z003 and Z014) and one sample was American bison with ancestral cattle DNA (Z011).

### 3.3. Mislabeled Products 

Based on the combined results of DNA barcoding and PCR-SFLP, 3 of the 45 samples (6.7%) tested in this study were determined to be mislabeled (Table 4). The mislabeled samples all consisted of ground meat—two were cooked burgers and one was a raw ground product. One sample (Z021) was advertised as a “Buffalo Burger” but was identified with DNA barcoding as red deer. This sample was purchased at a chain restaurant for USD 11.84. Along these lines, a game meats distributor in the US previously received a warning letter from the Food and Drug Administration (FDA) for selling “Black Bear Burgers” that instead contained elk/red deer (*Cervus* sp.) [26]. Similarly, Kane and Hellberg [18] previously reported the mislabeling of a ground bison product that was identified as containing American elk (*Cervus canadensis*), beef (*B. taurus*), pork (*Sus scrofa*), and horse (*Equus caballus*). The substitution of bison for *Cervus* species is likely economically motivated because bison meat can sell for higher prices than red deer/elk meat.

Two samples (Z003 and Z014) were labeled as bison but confirmed to contain domestic cattle (Table 4). Sample Z003 was a 0.45 kg packaged ground sample labeled as “Ground Bison, 90% Lean-10% Fat” and purchased from a warehouse-style grocery store for USD 9.99 (i.e., USD 22.20/kg). The average retail price of ground beef during the period this sample was purchased (July 2019) was USD 8.36/kg [14]. Therefore, sample Z003 was associated with an estimated economic profit of USD 13.84/kg. Sample Z014 was a 7-oz burger advertised as a “BBQ Bison Burger” and purchased from a local restaurant for USD 17.64. Comparatively, the price of a beef burger purchased from the same establishment was USD 15.00, resulting in an economic profit of USD 2.64 for this sample. In comparison, a previous study investigating whole cuts of game meat with DNA barcoding identified domestic cattle in one sample of bison stew meat and one sample of bison rib-eye steak [19], while Kane and Hellberg [18] detected domestic cattle in ground bison meat. Furthermore, a study on DNA barcoding of pet foods reported the detection of domestic cattle in a can of dog food labeled as containing buffalo [20]. However, previous studies utilized mitochondrial DNA targets and did not conduct supplementary PCR-SFLP testing to provide additional data as to whether the samples contained DNA from domestic cattle or hybridized bison. Similarly, previous studies also reported the identification of domestic cattle in products labeled as yak burgers [18] and yak steak [19]. However, due to historical crossbreeding of yak (*Bos grunniens*) and domestic cattle, these mislabeling events could not be confirmed. Therefore, follow-up testing is recommended for future studies on bison or yak products to verify the results of DNA barcoding.

When products were separated based on purchasing locations, mislabeling was detected in 2 of 21 samples (9.5%) purchased at restaurants and 1 of the 17 samples (5.9%) purchased at grocery stores. None of the four samples purchased from butchers or the three samples purchased online were mislabeled. In comparison, previous studies reported bison mislabeling in products purchased from online vendors. However, these studies only conducted sampling online or using a combination of online vendors and grocery stores. This was the first study to report DNA-based testing of bison products collected from restaurants.

The overall rate of mislabeling in the current study (6.7%) was relatively low compared to previous studies in North America that investigated a wider scope of meat products, with reported mislabeling rates of 14–21% [18,19,27,28]. However, it is difficult to make direct comparisons among mislabeling rates due to variation in sampling plans and species targeted by the different studies.

### 3.4. Additional Labeling Concerns

Eight of the 45 products collected in this study were labeled as “buffalo”—six cooked burger patties purchased from restaurants and two uncooked burger patties/ground meat purchased from grocery stores. Similarly, Kane and Hellberg [18] identified American bison in ground meat products labeled as “buffalo” and Hellberg et al. [20] identified American bison in two pet food products labeled as “buffalo jerky” and “buffalo patties.” Although the terms “buffalo” and “American buffalo” have historically been used to describe the American bison, the American bison is not a true species of buffalo and instead belongs to the *Bison* genus [29]. True buffalo species include the Asian water buffalo (*Bubalus bubalis*) and Cape buffalo (*Syncerus caffer caffer*), which are native to Asia and Africa, respectively. A previous study into game meats sold in the United States tested two samples of whole cuts labeled as “buffalo” and identified them as Asian water buffalo [19]. The use of the term “buffalo” to describe meat from the American bison may cause confusion, leading consumers to believe they are purchasing meat from true buffalo. To avoid deceptive labeling, it is recommended that only the term bison is used to describe meat from the American bison.

## 4. Conclusions

Overall, the results of this study revealed a relatively low level of mislabeling (6.7%) among the bison samples tested. The greatest mislabeling rate was found in samples purchased from restaurants (9.5%), followed by grocery stores (5.9%). No mislabeling was detected in samples purchased from butchers or online sources. The common trend of lower-cost species being substituted for higher-cost species for economic gain remains evident. This study demonstrated the importance of confirmation testing for bison products that test positive for domestic cattle with DNA barcoding. While three bison products in this study initially tested positive for domestic cattle with DNA barcoding, follow-up testing with PCR-SFLP indicated that one of the products was likely bison with ancestral cattle DNA. Further research should be conducted to examine the effectiveness of PCR-SFLP for the differentiation of other species with historical hybridization events, such as yak.

## Figures and Tables

**Figure 1 foods-10-00347-f001:**
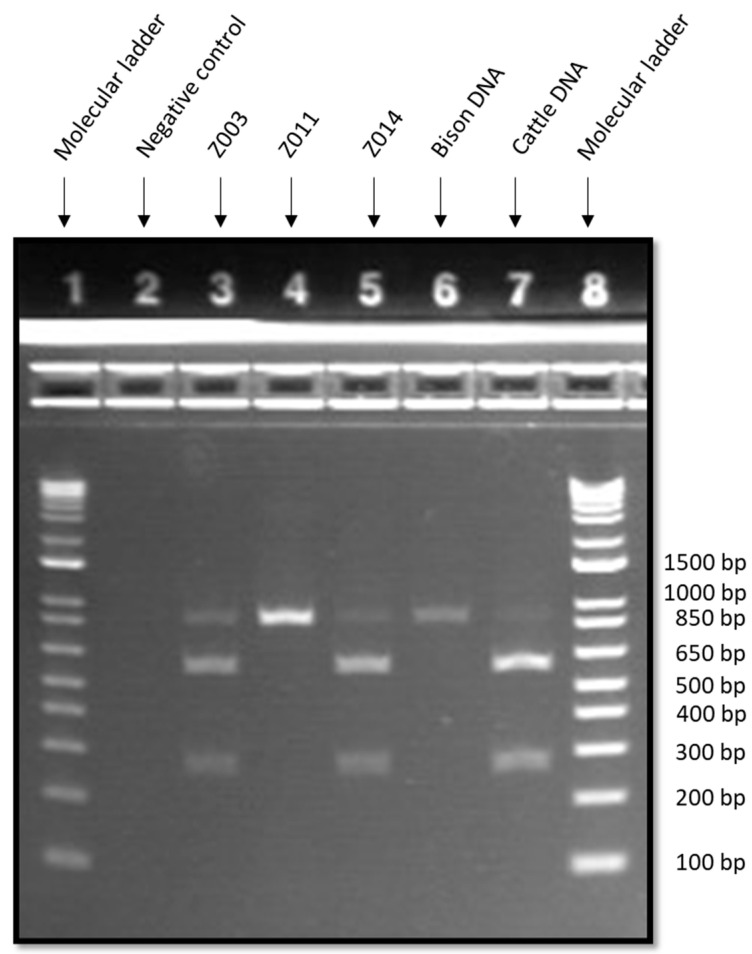
Gel electrophoresis results of polymerase chain reaction-satellite fragment length polymorphism (PCR-SFLP) (satellite 1.711 with *Taq*I digest) for bison products that were positive for domestic cattle with DNA barcoding. Well assignments are as follows: E-Gel^TM^ 1 Kb Plus DNA ladder (wells 1 and 8), non-template control (well 2), Z003 (well 3), Z011 (well 4), Z014 (well 5), bison DNA positive control (well 6), and domestic cattle DNA positive control (well 7).

**Table 1 foods-10-00347-t001:** Bison products tested in this study.

Collection Site	Number of Sites Visited	Number of Each Product Type Collected				Total Number of Products Collected
		Cooked Burgers	Uncooked Burger Patties/Ground Meat	Uncooked Whole Cuts/Steaks	Hot Dogs	
Grocery stores	9	0	11	6	0	17
Restaurants	21	21	0	0	0	21
Butchers	4	0	1	3	0	4
Online vendors	1	0	1	1	1	3
Overall	35	21	13	10	1	45

**Table 2 foods-10-00347-t002:** PCR primers used in this study.

Primer Set	Primer Name	Primer Direction	Primer Sequence (3′-5′) ^1^	Ratio in Cocktail	Product Length	Reference
Mammalian primer cocktail for DNA barcoding	LepF1_t1	Forward	TGTAAAACGACGGCCAGTATTCAACCAATCATAAAGATATTGG	1	658 bp ^2^	Ivanova et al. [23]
	VF1_t1	Forward	TGTAAAACGACGGCCAGTTCTCAACCAACCACAAAGACATTGG	1		
	VF1d_t1	Forward	TGTAAAACGACGGCCAGTTCTCAACCAACCACAA RGAYATYGG	1		
	VF1i_t1	Forward	TGTAAAACGACGGCCAGTTCTCAACCAACCAIAAI GAIATIGG	3		
	LepR1_t1	Reverse	CAGGAAACAGCTATGACTAAACTTCTGGATGTCC AAAAAATCA			
	VR1d_t1	Reverse	CAGGAAACAGCTATGACTAGACTTCTGGGTGGCC RAARAAYCA	1		
	VR1_t1	Reverse	CAGGAAACAGCTATGACTAGACTTCTGGGTGGCCAAAGAATCA	1		
	VR1i_t1	Reverse	CAGGAAACAGCTATGACTAGACTTCTGGGTGICCI AAIAAICA	3		
Satellite 1.711b	1.711b_F	Forward	CTGGGTGTGACAGTGTTTAAC	1	822 bp	Verkaar et al. [21]
	1.711b_R	Reverse	TGATCCAGGGTATTCGAAGGA	1		

^1^ Underlined segment indicates M13 tails. ^2^ DNA barcode product length does not include primers.

**Table 3 foods-10-00347-t003:** Results of PCR-SFLP for bison products (*n* = 3) that tested positive for domestic cattle with DNA barcoding. The satellite 1.711 PCR amplicon was digested with *Taq*I restriction enzyme to determine the species present in the sample.

Sample	Estimated Length of PCR-SFLP Fragments (bp)	Species Determination
Z003	257, 552, 809	Domestic cattle (*Bos taurus*)
Z011	822	American bison (*Bison bison*)
Z014	257, 552, 809	Domestic cattle (*Bos taurus*)
Bison DNA positive control	822	American bison (*Bison bison*)
Cattle DNA positive control	257, 552, 809 (faint)	Domestic cattle (*Bos taurus*)
Non-template control	No fragments	Negative

**Table 4 foods-10-00347-t004:** Summary of mislabeled bison products identified in this study.

Sample ID	Product Description on Label/Menu	Purchase Location	Price Paid (USD)	Amount Purchased	Product Type	Identified Species
Z021	Buffalo burger	Restaurant	$11.84	1 burger (no weight given)	Cooked burger	Red deer (*Cervus elaphus*)
Z003	Fresh 90/10 ground bison	Grocery store	$9.99	0.45 kg prepackaged	Uncooked ground	Domestic cattle (*Bos taurus*)
Z014	Bison burger	Restaurant	$17.64	1 burger (no weight given)	Cooked burger	Domestic cattle (*Bos taurus*)

## Data Availability

The data presented in this study are available in Appendix A.

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
