# Peer review of "Use of DNA Barcoding Combined with PCR-SFLP to Authenticate Species in Bison Meat Products"

_foods, 2021, doi:10.3390/foods10020347_

Round 1
Reviewer 1 Report
General comments
This study uses DNA data to examine the identity of meat products labelled as bison. Previous studies have examined bison meat products using only mtDNA sequencing. This study adds a second locus from the nuclear genome to provide additional confirmation about the identity of the products, which is important given that bison and cattle have hybridized in the past and therefore may share some genetic markers. This study is well written (although a few more details in the methods would be useful) and on an interesting topic.
Specific comments
Ln 19: The final sentence of the abstract could be rewritten to read: This study revealed that additional DNA testing of species that have undergone historical hybridization provides improved identification results compared to DNA barcoding alone.
Ln 66: Change (PCR) to (RT-PCR).
Ln 140: Were the primer cocktails used for sequencing? If so, were the proportions of each primer in the same ratio as given in Table 2? Or did you use the M13 tag primer for sequencing?
Ln 147: Why were sequences with >2% ambiguities discarded? These may represent samples containing more than one species (such as mince made from a mix of bison and cattle).
Ln 159: Change to ‘15 s of 92°C’
Table 2: If the ‘forward’ cocktails and ‘reverse’ cocktails are used for sequencing then it needs to be clear in this table which primer goes in which cocktail (I assume that the first four primers are in the forward cocktail?). An extra column could record this information.
Ln 187: I think it would be a good idea to give the names of the 3 samples (Z003, Z011, Z014) identified as domestic cattle here, since they are referred to by name in the next section.
Ln 214: delete the word bison from ‘samples consisted of ground bison meat’ since they were found to be not bison!
Ln 243: rather than write ‘confirm whether the samples contained DNA from domestic cattle or hybridized bison’ I would suggest rewriting as ‘provide additional data as to whether the samples contained DNA from domestic cattle or hybridized bison’. The satellite DNA is only a single additional locus so there will still be (rare) instances when a sample has a bison mtDNA and bison satellite DNA but have cattle ancestry.
Ln 271: Bison genus should be Bison genus.
Ln 277: it would be clearer to state ‘it is recommended that only the term bison is used….”
References: there is some inconsistency over whether the article titles are all in lower case or not e.g. compare references 2 and 4.
Ln 381: Change Availabe to Available.
Author Response
Response to Reviewer 1
This study uses DNA data to examine the identity of meat products labelled as bison. Previous studies have examined bison meat products using only mtDNA sequencing. This study adds a second locus from the nuclear genome to provide additional confirmation about the identity of the products, which is important given that bison and cattle have hybridized in the past and therefore may share some genetic markers. This study is well written (although a few more details in the methods would be useful) and on an interesting topic.
Author Response: Thank you for your helpful review of our manuscript. We have provided additional details in the methods section, as suggested in the comments below.
Specific comments
Ln 19: The final sentence of the abstract could be rewritten to read: This study revealed that additional DNA testing of species that have undergone historical hybridization provides improved identification results compared to DNA barcoding alone.
Author Response: Thank you for the suggestions. We have revised the sentence as recommended.
Ln 66: Change (PCR) to (RT-PCR).
Author Response: The acronym RT-PCR is commonly used to describe reverse-transcription PCR. As stated in The MIQE Guidelines: Minimum Information for Publication of Quantitative Real-Time PCR Experiments (https://doi.org/10.1373/clinchem.2008.112797), use of this acronym to describe real-time PCR can cause confusion. Real-time PCR assays that are quantitative can be referred to using the acronym “qPCR”. However, not all real-time PCR assays targeting meat species identification are quantitative. Therefore, we decided to change “(PCR)” to “(real-time PCR)”.
Ln 140: Were the primer cocktails used for sequencing? If so, were the proportions of each primer in the same ratio as given in Table 2? Or did you use the M13 tag primer for sequencing?
Author Response: The M13 tails shown in Table 2 were used for DNA sequencing. This detail has been added to the description of DNA sequencing in the methods (line 208).
Ln 147: Why were sequences with >2% ambiguities discarded? These may represent samples containing more than one species (such as mince made from a mix of bison and cattle).
Author Response: All of the sequences had <2% ambiguities, so none were discarded. This information has been added to the results section (lines 337-338). To avoid confusion, we removed the statement about <2% ambiguities from the methods section.
Ln 159: Change to ‘15 s of 92°C’
Author Response: We changed the text to read “15 s at 92 ℃” to be consistent with the wording used in the rest of the sentence.
Table 2: If the ‘forward’ cocktails and ‘reverse’ cocktails are used for sequencing then it needs to be clear in this table which primer goes in which cocktail (I assume that the first four primers are in the forward cocktail?). An extra column could record this information.
Author Response: As described above, the M13 tails were used for DNA sequencing and this has been clarified in the manuscript. We added an extra column to Table 2 that describes the primer direction (forward or reverse).
Ln 187: I think it would be a good idea to give the names of the 3 samples (Z003, Z011, Z014) identified as domestic cattle here, since they are referred to by name in the next section.
Author Response: Thank you for the suggestion. We added the sample IDs to this sentence.
Ln 214: delete the word bison from ‘samples consisted of ground bison meat’ since they were found to be not bison!
Author Response: Thank you for catching this. We have made the suggested correction.
Ln 243: rather than write ‘confirm whether the samples contained DNA from domestic cattle or hybridized bison’ I would suggest rewriting as ‘provide additional data as to whether the samples contained DNA from domestic cattle or hybridized bison’. The satellite DNA is only a single additional locus so there will still be (rare) instances when a sample has a bison mtDNA and bison satellite DNA but have cattle ancestry.
Author Response: This is an excellent point and we have made the suggested correction.
Ln 271: Bison genus should be Bison genus.
Author Response: Correction has been made.
Ln 277: it would be clearer to state ‘it is recommended that only the term bison is used….”
Author Response: Correction has been made.
References: there is some inconsistency over whether the article titles are all in lower case or not e.g. compare references 2 and 4.
Author Response: We have adjusted the article titles to be in sentence case.
Ln 381: Change Availabe to Available.
Author Response: Correction has been made.
Reviewer 2 Report
The work presented is clear in its purpose, that is the recognition of america bison in meat products sold in the market, and reliable in its results obtained by a combination of DNA barcoding and PCR-SFLP approaches, this latter applied to those products suspected to result from substitutions. The paper is very simple because the task is very focused. Results are convincing although a better gel for Fig.1 is higly recommended because the difference between the fragments of 822 (bison) and 809 (cattle) is not appreciable and, moreover, the 809 bp band in the positive cattle control is fainter that in the two product samples (Z003 , Z014) identifed as undeclared substitutions. Overall, this paper deliver clear, limited but overly discussed results. In fact, sections 3.3 and 3.4 keep on delivering the same repetitive information. In my opinion, they should be shortened.
Author Response
The work presented is clear in its purpose, that is the recognition of america bison in meat products sold in the market, and reliable in its results obtained by a combination of DNA barcoding and PCR-SFLP approaches, this latter applied to those products suspected to result from substitutions. The paper is very simple because the task is very focused. Results are convincing although a better gel for Fig.1 is higly recommended because the difference between the fragments of 822 (bison) and 809 (cattle) is not appreciable and, moreover, the 809 bp band in the positive cattle control is fainter that in the two product samples (Z003 , Z014) identifed as undeclared substitutions. Overall, this paper deliver clear, limited but overly discussed results. In fact, sections 3.3 and 3.4 keep on delivering the same repetitive information. In my opinion, they should be shortened.
Author Response: Thank you for your review of our manuscript. We respectfully disagree with the need to re-run the gel for Fig. 1. Although there is not a notable difference between fragments of 809 bp (cattle) and 822 bp (bison) in Fig. 1, the overall banding pattern for each species is clearly different and enables identification of species. The 809 bp band for the cattle is expected to be relatively faint because the fragment has been cleaved by the restriction enzyme into shorter fragments of 257 and 552 bp. It is the cleavage of the larger fragment into the two shorter fragments that allows for the differentiation of cattle and bison. With regards to Sections 3.3 and 3.4, Section 3.3 has been shortened to avoid repetition. Section 3.4 covers a slightly different topic, which is bison products that are labeled as buffalo. While some of the same studies are cited in both sections, the labeling/mislabeling events that are discussed are different.